# Surgical Management of Pancreatic Neuroendocrine Tumors

**DOI:** 10.3390/cancers15072006

**Published:** 2023-03-28

**Authors:** Megan L. Sulciner, Thomas E. Clancy

**Affiliations:** Division of Surgical Oncology, Department of Surgery, Brigham and Women’s Hospital, Boston, MA 02115, USA

**Keywords:** pancreatic neuroendocrine tumors, small nonfunctional, surgery, metastasis, observation

## Abstract

**Simple Summary:**

This review, as part of a series of reviews on neuroendocrine tumors, focuses on the particular management strategies for pancreatic neuroendocrine tumors. While far less common than pancreatic ductal adenocarcinoma, neuroendocrine tumors of the pancreas are increasingly recognized in the setting of high-quality cross-sectional imaging. These tumors demonstrate a range of behavior from the nonfunctional to hormone-secreting functional tumors, and from relatively indolent neoplasms to those with more aggressive behavior. Management principles unique to these tumors are addressed, including the role of surgery for both oncologic as well as palliative goals, indications for surgery versus observation in small nonfunctional tumors, and management of metastatic disease.

**Abstract:**

Pancreatic neuroendocrine tumors (PNETs) are relatively uncommon malignancies, characterized as either functional or nonfunctional secondary to their secretion of biologically active hormones. A wide range of clinical behavior can be seen, with the primary prognostic indicator being tumor grade as defined by the Ki67 proliferation index and mitotic index. Surgery is the primary treatment modality for PNETs. While functional PNETs should undergo resection for symptom control as well as potential curative intent, nonfunctional PNETs are increasingly managed nonoperatively. There is increasing data to suggest small, nonfunctional PNETs (less than 2 cm) are appropriate follow with nonoperative active surveillance. Evidence supports surgical management of metastatic disease if possible, and occasionally even surgical management of the primary tumor in the setting of widespread metastases. In this review, we highlight the evolving surgical management of local and metastatic PNETs.

## 1. Introduction

Pancreatic neuroendocrine tumors (PNETs) are rare malignancies originating from the endocrine tissues of the pancreas. PNETs represent approximately 2% of all pancreatic neoplasms, yet the overall incidence appears to be increasing [1,2,3]. This observation is perhaps a result of increased recognition in the setting of improved quality of cross-sectional imaging. PNETs are mainly sporadic, though approximately 10% occur in the context of genetic predispositions including multiple endocrine neoplasia type 1 (MEN1), von Hippel Lindau (VHL) disease, tuberous sclerosis complex (TSC1 and TSC2), and neurofibromatosis (NF1) [4]. PNETs can present with a range of clinical behaviors; they may occur as small benign lesions, slow-growing indolent tumors with a favorable prognosis, locally invasive lesions, or as widespread metastatic disease. PNETs are categorized as either functional or nonfunctional based on the ability to secrete biologically active hormones. Functional PNETs secrete specific hormones with subsequent characteristic symptoms and are far less prevalent than nonfunctional PNETs. Nonfunctional lesions have previously thought to comprise approximately 50–85% of all PNETs [4,5], though the frequency is likely greater given increased recognition of small, indolent lesions. Overall, the prevalence of nonfunctioning PNETs is on the rise, specifically an increased incidence of small nonfunctional PNETs (less than 2 cm) [1,6,7].

While tumor stage is based on the American Joint Committee on Cancer (AJCC) TNM staging system, the main prognostic determinant for PNETs is tumor grade [8]. Tumor grade is determined by immunohistochemistry analysis of tissues obtained via biopsy [9,10,11]. Well differentiated neuroendocrine tumors are assigned a grade 1–3 based on the Ki67 proliferation index and mitotic index per high power field [8,12]. Ki67, a nuclear protein associated with proliferation, is quantified in tumor tissue using immunohistochemical staining with MIB1, an antibody against Ki67. High Ki67, or increased staining with MIB1, has been correlated with worse outcomes. Further increased expression of Ki67 in PNETs is considered a high-risk feature. Mitotic index is an additional determinant of tumor grade. Grade 1 tumors are characterized by <2 mitoses/2 mm^2^ and/or Ki67 <3%; Grade 2 tumors by 2–20 mitoses/2 mm^2^ and/or Ki67 3–20%; and Grade 3 tumors by >20 mitoses/2 mm^2^ and/or Ki67 >20% [12,13] (Table 1). Alternatively, poorly differentiated PNETs are considered high grade via >20 mitoses/2 mm^2^ and/or Ki67 >20%, but are further classified based on cell size [12,13]. Median overall survival for patients with grade 1 PNETs is 12 years compared to grade 3 disease with a median survival of 10 months [1].

While histologic grade remains the mainstay of PNET prognosis, alternative prognostic markers is an area of ongoing investigation. Larger tumor size (>2 cm), symptomatic tumors, Ki67 >3%, and positive lymph nodes have also been correlated with increased risk of recurrence [14]. Additionally, tissue diagnosis is often obtained by endoscopic ultrasound with fine-needle aspiration (FNA) for patients with PNETs less than 2 cm; however, diagnostic yield from FNA has been poor. A recent study by the US Neuroendocrine Tumor Study Group demonstrated that in an endoscopic ultrasound FNA on a lesion less than 2 cm, tumor differentiation and Ki67 index could only be determined in 26.4% and 20.1% of cases, respectively [15]. Thus, if a suspicious lesion is unable to be biopsied, diagnosis relies on somatostatin-receptor imaging, even for small, non-functional PNETs [9,16]. Utilization of next-generation sequencing may also be on the horizon for improved stratification [17].

Surgical resection remains the primary curative-intent modality for localized PNET [4], and resection may be associated with improved oncologic outcomes even in some cases of metastatic disease. In addition to oncologic benefits, tumor resection may alleviate symptoms secondary to the hormone secretion for functional PNETs, and resection of primary functioning PNETs is associated with improved overall survival for all stages of disease [18]. The variable behavior of these tumors precludes a single surgical strategy, with management options ranging from nonoperative observation to pancreas-sparing procedures, pancreatectomy with lymphadenectomy, and occasionally metastasectomy. Herein, we discuss the presentation, diagnosis, and surgical management for both functioning and nonfunctioning PNETs, as well as new treatment directions.

## 2. Presentation and Diagnosis of Pancreatic Neuroendocrine Tumors

Many patients with nonfunctional PNET will be asymptomatic with incidental imaging findings, and rarely present with symptoms from a tumor mass effect. Symptoms may be present with functional tumors, based on the specific hormone secreted. If a PNET is suspected, multiphasic CT with intravenous contrast is preferred [19]. PNETs are characteristically hyperenhancing on the arterial phase, secondary to the hypervascularity of the lesions. On CT, PNETs frequently present as a solid or cystic hyperenhancing mass, occasionally with associated calcification (Figure 1). Detection on CT is size-dependent, with sensitivity of approximately 82% and specificity of 96% [20,21]. 

MRI is also useful in the diagnosis of PNETs with detection sensitivity of 93% and specificity of 88% [19]. MRI is particularly useful in the characterization of liver metastases [22]. MRI may also assist in disease stratification, as tumors with distinct MRI features have been correlated with tumor aggressiveness and progression-free survival after tumor resection. In a small, retrospective study of 80 patients, aggressive tumors were defined as greater than 2 cm; T2 non-bright lesions on T2 weighted images had pancreatic duct dilation, and restricted diffusion within the lesion [23].

Nuclear imaging techniques are particularly sensitive for detecting many PNETs. Most well-differentiated PNET express the somatostatin receptor (SSTR), a feature which may be exploited using radiolabeled somatostatin analogs. 68Ga-DOTATATE and DOTANOC are functional imaging radiopharmaceuticals using the positron-emitting radioisotope gallium-68. (Figure 2) This allows the use of PET/CT imaging to detect SSTR-expressing PNET with sensitivity of 93% and specificity of 91% [24]. 68Ga-DOTATATE PET/CT has demonstrated efficacy in the detection of primarily low-grade tumors, G1 and G2 PNETs, with detection rates that have been reported as 95% and 87.5%, respectively [25]. However, for higher grade tumors (G3), this imaging modality is significantly less effective, with 37.5% detected [25]. An additional work-up should be pursued in patients with high clinical suspicion and negative DOTATATE scan, such as traditional radiolabeled fluorodeoxyglucose (FDG) PET imaging.

The most sensitive method for diagnosing PNETs is endoscopic ultrasound (EUS), with a mean sensitivity range of 75–97% [21]. EUS is particularly sensitive for PNETs that are less than 2 cm [26,27]. In fact, in a recent meta-analysis, EUS demonstrated increased rates of PNET detection even after other imaging modalities failed to identify a lesion [28]. Intraoperative ultrasound may serve as an adjunct to preoperative imaging. Intraoperative ultrasound can be useful for detecting smaller PNETs with reports of a detection rate of up to 96% [21,29]. In fact, when combined with palpation, intraoperative ultrasound detected all tumors in a small, single-institution retrospective study [29].

Intraductal ultrasound (IDUS) utilizes miniprobes that can be passed directly into pancreatic and bile ducts through a standard endoscope. IDUS has been most often used in the diagnosis and investigation of biliary disease; however, there is limited data to suggest IDUS may be an alternative diagnostic modality for PNETs [30,31]. In one study, three patients with presumed PNETs underwent several imaging modalities, including endoscopic ultrasound, without tumor localization. All patients were found to have PNETs less than 2 cm and were detected by IDUS [30].

Nonfunctional PNETs comprise the majority of all pancreatic neuroendocrine tumors. Nonfunctional PNETs characteristically do not secrete hormones, with most remaining asymptomatic. For those nonfunctional PNETs causing symptoms, presentation is more often due to large tumor burden causing mass effect on nearby structures.

Presentation of functional PNET varies widely depending on the hormones secreted. Insulinomas are functional PNETs that arise from the pancreatic beta cells, resulting in unregulated secretion of insulin. These lesions are typically small and evenly distributed throughout the pancreas. Most are benign, with less than 8% being malignant [32,33]. Diagnosis of insulinomas relies on clinician suspicion, as patients typically present with symptoms of hypoglycemia, including neurologic (fatigue, mental status change, headache, or blurred vision) or sympathetic (palpitations, sweating, or tremors) related symptoms. Additional laboratory studies for the diagnosis of insulinomas includes quantitation of insulin, glucose, C peptide, and proinsulin. Tumors are often symptomatic at a small size, and typical imaging modalities may fail to recognize them. Endoscopic ultrasound may be useful for detecting small tumors, having a mean detection rate of 86% [21], with the most accurate detection occurring with CT followed in sequence by endoscopic ultrasound [27]. MRI has a sensitivity of 85% for the detection of insulinomas [34]. Somatostatin receptor scintigraphy has more limited use for the detection of insulinomas given the low expression of the somatostatin receptor [35]. Rarely, for small occult lesions, localization of the lesion to a general area of the pancreas can be achieved with selective arterial calcium stimulation angiogram [36].

Gastrinomas can be multifocal within the abdomen; however, up to 90% of these tumors are present within the “gastrinoma triangle”, the junction of the second and third part of the duodenum, the cystic duct and common bile duct junction, and the body and neck of the pancreas junction [37]. Patients with gastrinomas may present with multiple or recurrent upper gastrointestinal ulcers. Excess gastrin secretion can present as Zollinger-Ellison syndrome, first described in 1955 in two cases of patients with jejunal ulcers who were found to have gastrin hypersecretion [38]. Additional presenting symptoms of a gastrinoma include weight loss and other complications of high acid secretion such as bleeding, stricture, fistulas, or visceral perforations. Occasionally, gastrinomas may be associated with “type II” gastric neuroendocrine tumors related to hypergastrinemia. Diagnosis of gastrinomas is a combination of clinical suspicion based on symptom presentation, in addition to laboratory and imaging studies. Laboratories can include fasting serum gastrin level in combination with gastric pH level. Localization may be challenging with CT or MRI; thus, somatostatin receptor scintigraphy may be used for tumor localization. If patient is presenting symptoms and laboratories are suggestive of gastrinoma, though imaging is negative, endoscopic ultrasound may be considered, given its sensitivity for small tumors [27].

Additional functional PNETs are even less commonly encountered. Glucagonomas arise from the alpha islet cells located in the distal pancreas. The presenting symptoms of glucagonomas are secondary to their secretion of glucagon, with associated diabetes, weight loss, and characteristic rash of necrolytic migratory erythema. The diagnosis of a glucagonoma requires high clinical suspicion and then is confirmed via elevated serum glucagon, a glucose level greater than 500 pg/mL, and imaging studies [39]. VIPomas may present with watery diarrhea, hypokalemia, and achlorhydria, the “WDHA” or Verner-Morrison syndrome [40]. These symptoms are secondary to VIPomas’ hypersecretion of vasoactive intestinal peptide (VIP). VIPomas often arise in the pancreatic tail and are often large and metastatic at presentation. Diagnosis of VIPomas is similar to the previously described functional PNETs, including a high clinical suspicion based on symptoms and characteristic laboratory findings. For patients with VIPomas, laboratory studies may reveal hypokalemia and other electrolyte derangements secondary to dehydration in the setting of watery diarrhea. Somatostatinomas originate from the delta cells of the pancreas. Presenting symptoms are secondary to the inhibitory effects of somatostatin, thus patients may present with cholelithiasis, diabetes, and/or steatorrhea. These tumors often present with late-stage disease.

In addition to the well-characterized functional PNETs, there are several very rare PNETs that secrete other hormones and signaling peptides [41]. PNETs secreting serotonin comprise up to 4% of all PNETs [42]. Carcinoid syndrome is a collection of symptoms that arises due to secretion of serotonin or tachykinins. The most common symptoms of carcinoid syndrome include flushing and diarrhea, as well as other symptoms secondary to persistent vasodilation. However, carcinoid syndrome occurs only in the minority of all patients with serotonin secreting PNETs [42].

Parathyroid hormone-related peptide (PTHrP) is produced by a range of cancer types, including PNETs, though secretion of PTHrP from metastatic PNETs is incredibly rare and can result in hypercalcemia [43]. There have also been case series describing PNETs secreting renin and erythropoietin, resulting in hypertension and polycythemia, respectively [44,45]. Cholecystokinin (CCK) is a peptide hormone responsible for stimulating the release of pancreatic enzymes and the processes that result in bile secretion into the duodenum, such as gallbladder contraction and relaxation of the sphincter of Oddi. CCK secretion by a PNET is also very rare, though it can present with symptoms similar to Zollinger-Ellison syndrome such as a peptic ulcer, in addition to weight loss and diarrhea [46]. Similarly, there have been case series in which patients presented with hypoglycemic symptoms and were found to have PNET secretion IGF-2 or GLP-1 [47,48].

Luteinizing hormone (LH), secreted by the pituitary gland, targets reproductive organs to then secrete androgens. However, another very rare PNET that secretes LH has been described. In one case study, a 61-year-old man presenting with generalized fatigue and erectile dysfunction was found to have an elevated LH level. CT imaging revealed a mass in the pancreatic tail, for which he received a resection and subsequent normalization of his LH [49]. PNET secretion of growth-hormone releasing hormone (GHRH) and adrenocorticotropic hormone (ACTH) have been reported in the literature [50,51,52], with patients presenting with the characteristic symptoms of acromegaly and Cushing syndrome, respectively.

## 3. Surgical Management of Pancreatic Neuroendocrine Tumors

Surgery is the main modality to achieve locoregional control for well-differentiated low-grade PNET, and the only potential cure for PNETs. Median overall survival is improved with resection for all low-grade PNETs, regardless of functional status [1,53]. The role of surgical resection in localized high-grade well-differentiated PNETs is more limited, given their much more aggressive behavior [54], with surgery not indicated for poorly differentiated lesions. Despite differences in presentation, surgical approaches to functional and nonfunctional tumors are generally similar, generally consisting of partial pancreatectomy via traditional open or minimally invasive approaches. Operative approaches includes both open and minimally invasive techniques. Compared to traditional open approaches, laparoscopic distal pancreatectomy has been associated with a reduction in postoperative morbidity, with similar oncologic and survival outcomes [55]. Prospective data also supports the use of minimally invasive surgery when possible, particularly for distal pancreatic tumors [56]. Particular considerations for various PNETs are discussed below, with attention to specific issues with different functional PNETs, increased use of nonoperative therapy for small nonfunctional PNETs, the role of lymphadenectomy, the role of pancreas-sparing enucleation, and the role of metastasectomy.

### 3.1. Functional PNETs

All localized, functional PNETs should be considered for surgical resection [57]. Resection for functional tumors provides source control for symptoms secondary to hormone secretion, in addition to curative-intent treatment for the prevention of metastases.

Surgical management of insulinomas can often be achieved via pancreas-sparing procedures [58]. Given symptoms, surgical exploration should be considered for all patients with resectable disease regardless of size after efforts at tumor localization [59]. Preoperative localization may require the use of endoscopic ultrasound in addition to cross-sectional imaging. In the particular case of MEN1, due to a high rate of multifocal disease, it is important to identify insulin-producing lesions preoperatively in an effort to spare pancreatic parenchyma, as all PNETs may not be insulin-secreting [60]. The majority of insulinomas are benign [59], and extensive lymph node dissection is not indicated. As a result, parenchyma-preserving enucleation is preferred if possible. Enucleation involves removal of just the tumor and associated capsule, otherwise sparing normal parenchyma (Figure 3). This is particularly applicable to small, benign, superficial tumors, but less possible for tumors with abutment of the main pancreatic duct or deeper in the pancreatic parenchyma. This approach is associated with improved postoperative outcomes compared to resection in patients with small PNETs (mean 2.3 cm) [61]. Minimally invasive enucleation has also been associated with lower perioperative morbidity than the open approach [58]. However, enucleation has also been associated with is an increased risk of postoperative pancreatic fistula formation, and careful attention must be given to the relationship between an insulinoma and the pancreatic duct [53]. Implementation of laparoscopic ultrasound has been useful for intraoperative localization, especially for smaller insulinomas, multiple lesion exclusion, and identification of the main pancreatic duct [62].

Prior to current localization techniques, pursuit of blind distal pancreatectomy had been advocated lesions that had not been identified on imaging or during abdominal exploration [63]. Currently, this is rarely indicated given improved success with preoperative localization. In rare cases, localization of small occult lesions to a general area of the pancreas (i.e., tail, body, head) with the use selective arterial calcium stimulation angiogram may be needed to guide resection (Figure 4).

Patients with insulinomas deemed to be poor surgical candidates may benefit from endoscopic or percutaneous ablative therapy. Limited data suggests that endoscopic ultrasound-guided radiofrequency ablation may be feasible with small insulinomas. This modality induces coagulative necrosis via directed thermal energy, and it has been used in the treatment of other solid tumors. Several studies have demonstrated this technique is safe and with minimal post-procedure complications [64,65]. From a technical standpoint, clinical success appears to be higher for insulinomas that are located in head or neck of the pancreas [65]. While studies are limited, endoscopic ultrasound-guided radiofrequency has demonstrated symptomatic improvement after radiofrequency [64]. Additional studies are needed, but minimally invasive techniques for the treatment of insulinomas are feasible and may provide benefit by way of sparing healthy pancreatic tissue.

Other functional PNETs have some specific surgical considerations worth noting. Approximately 70% of pancreatic gastrinomas are malignant [66,67]. Resection is indicated not only for curative intent, but also for symptom management to reduce excess gastric acid. However, symptoms of hyperacidity and peptic ulcer disease can typically be managed with proton pump inhibitor (PPI) therapy and somatostatin analog therapy. Older data supported a policy of surgical exploration of the “gastrinoma triangle”, occasionally including duodenotomy, which may be required, though conventional practice would be to operate only on tumors localized with high-quality imaging. While small tumors in the setting of MEN1 might be observed, sporadic gastrinomas should undergo surgical exploration, unless there is a medical contraindication to surgery. Lymphadenectomy should be performed for prognostic information.

Management of less common functional PNETs is challenging to standardize based on their relatively rare incidence. As the majority of glucagonomas are malignant, surgical resection is indicated for curative intent. However, these lesions often present as large, bulky tumors rendering surgical resection difficult [68]. VIPomas require surgical resection for potential cure [69], though they are often large and metastatic on presentation [68]; therefore, complete resection is difficult to achieve. Reduction of tumor burden reduction (debulking) may be considered for symptomatic control if feasible [68]. The majority of somatostatinomas are malignant [70,71], and localized disease should undergo resection [69,72]. Smaller somatostatinomas (less than 2 cm) can be addressed via formal surgical resection or enucleation [72]. Curative-intent resection likely requires a pancreaticoduodenectomy, given the proclivity of the tumor arising in the head of the pancreas [69]. Meta-analysis on resection of somatostatinoma metastases has demonstrated no survival benefit [72]. Similar to the surgical management of glucagonoma and VIPomas, the data specific to somatostatinomas is lacking and reliant on other functional PNET studies.

### 3.2. Nonfunctional PNETs

Nonfunctional PNETs comprise the majority of all PNETs [70]. Nonfunctional PNETs have a range of malignant potential, in that these tumors can be slow and insidious, and also locally invasive or aggressively metastatic [8]. Nonfunctional PNETs may metastasize to the liver, bone, peritoneum, adrenal gland, brain, and spleen. The 5-year overall survival for nonfunctional PNETs has been reported to be 26–58% [73,74]. Given that nonfunctional PNETs do not secrete hormones, they often are found incidentally and may therefore present with advanced disease [70,75].

Unlike functional PNETs, the primary reason to resect asymptomatic nonfunctional PNETs is to prevent growth, spread, and impacts on patient survival. Given that their biologic behavior may range from slow-growing and indolent to aggressive with the potential to metastasize, a uniform approach is not obvious. A correlation between tumor size and risk of malignant characteristics has been demonstrated [76]. Nonfunctional PNETs that are symptomatic, large (greater than 2 cm), and with atypical features such as pancreatic duct dilatation, should undergo surgical resection [77]. However, the management of small nonfunctioning tumors has been an area of debate given their typically relatively indolent behavior. While additional tumor characteristics such as Ki-67 proliferative index could ideally guide decision making regarding resection [78], at present tumor size has been the most reliable determinant of tumor progression for well-differentiated PNET.

A National Cancer Database (NCDB) study of over 2000 PNET patients demonstrated active surveillance can be reasonably pursued for tumors less than 1 cm [79]. Similarly a surveillance, epidemiology, and end results (SEER) database study found the likelihood of aggressive behavior in nonfunctioning PNETs less than or equal to 2 cm was low, and there was no survival benefit with resection [80]. Several retrospective database studies evaluating the role of active surveillance for small (less than or equal to 2 cm) nonfunctional PNETs suggest that a nonoperative approach is safe [81,82,83,84,85,86,87] (Table 2). Importantly, all patients followed nonoperatively require definitive diagnosis via either fine needle aspiration (FNA) or somatostatin-receptor imaging. In these introspective studies, with patients followed with serial imaging for incidental nonfunctioning PNET, the vast majority of patients do not demonstrate tumor growth, nor do most patients develop disease-specific morbidity. The nonoperative strategy is beneficial primarily to avoid the known incidence of morbidity from surgical exploration of up to 62%, mainly due to pancreatic fistula [83]. One potential criticism of such retrospective series is the relatively short-term follow-up given a disease process that may evolve over decades. A more aggressive surgical posture has been advocated by some who suggest a rate of late metastases or recurrence patients with incidental nonfunctioning PNETs, including nearly 8% rate of recurrence or metastasis in patients with tumors 2 cm or smaller [88].

Randomized data to guide management of small, incidentally discovered PNETs are difficult to obtain due to their low incidence and relatively indolent course. A recent international, prospective, nonrandomized study attempts to address this, enrolling patients with asymptomatic small nonfunctional PNETs (2 cm or less) [89]. In an interim analysis, only 18.8% of included patients underwent surgical resection, with the majority pursuing active surveillance. The decision to pursue surgical resection for these patients included patient preference, younger age, tumor size greater than 1 cm, and presence of main pancreatic duct dilation [90]. At last follow-up, 2% of patients had tumor progression and no patients had metastatic disease [90]. In a single-institution study inclusive of 177 patients, tumor size greater than 2 cm was found to be an independent predictor of malignancy. Further, of patients with lesions less than 2 cm that were incidentally diagnosed, none were deceased at last follow-up due to disease [76]. Another single-institution, retrospective cohort study inclusive of 174 patients found the ENETS guidelines specifically for active surveillance in patients with low-grade PNETs that were less than 2 cm were a valid treatment strategy [91]. These preliminary results provide additional, prospective evidence that active surveillance is a safe strategy for patients with small nonfunctional PNETs.

While data remain limited, there are several expert society guidelines that have addressed the management of small, nonfunctional PNETs. Per a recent consensus statement by the North American Neuroendocrine Tumor Society (NANETS), asymptomatic patients with tumors less than 1 cm and imaging consistent with PNET can be observed. However, for tumors 1–2 cm, it is advised that management be based on patient comorbidities, tumor grade, extent of resection if surgery is pursued, patient preference, and access to follow-up care [57]. The National Comprehensive Cancer Network (NCCN) recommendations suggest that tumors less than or equal to 2 cm can be observed; however, evidence is stronger for surveillance of tumors less than or equal to 1 cm [11]. The Canadian National Expert Group consensus on nonfunctional PNET surgical management states for tumors less than 2 cm, active surveillance can reasonably be pursued; specifically, tumors should be solitary lesions with no evidence of invasive disease, have low Ki67, and continue to demonstrate stability on serial imaging and biochemical monitoring every 6 months [9]. The European Neuroendocrine Tumor Society (ENETS) recommends active surveillance for nonfunctional PNETs that are less than or equal to 2 cm [59]. Tumors should be low-grade (G2 or less), asymptomatic, and without radiographic evidence suspicious for malignancy. However, surgery is recommended if the tumor is symptomatic or patient’s preference is for resection. Active surveillance includes imaging every 6 to 12 months [59]. These consensus recommendations are summarized in Table 3.

Pertinent to the decision making around the pursuit of active surveillance as a management strategy is the significant morbidity associated with pancreatic resection [92,93,94,95]. Depending on extent and location of pancreatic resection, rates of pancreatic fistulas have been reported in up to 17% of patients [92,95,96]. Though notable, distal pancreatectomies have a pancreatic fistula rate of approximately 5%. Additionally, new-onset diabetes has been reported in up to 10% of all patients undergoing pancreatic resection [92,93] and exocrine insufficiency in as many as 50% of patients undergoing pancreaticoduodenectomy [93,95]. Further, a recent study found up 67% of patients who underwent pancreatic resection had postoperative gastrointestinal symptoms, with 1% of patients being diagnosed with small intestinal bacterial overgrowth [97].

The extent of resection required for nonfunctional PNET is also a topic of debate. Minimally invasive approaches, specifically laparoscopic and robotic, for abdominal surgeries, including pancreatic resection, are becoming increasingly prevalent [98,99]. In a large, multi-center retrospective study conducted by the Pancreatic Neuroendocrine Disease Alliance (PANDA), minimally invasive pancreatectomy for PNETs had improved perioperative and postoperative outcomes, with no difference in recurrence-free or overall survival [100]. This study included both laparoscopic and robotic approaches in the minimally invasive group. Additionally, pancreatic enucleation, commonly used for small benign insulinomas, has been suggested in an effort to avoid postoperative pancreatic insufficiency, particularly for tumors not in proximity to the pancreatic duct. The primary argument against the use of enucleation is the metastatic potential of nonfunctioning PNET. Enucleation will not provide staging information in the absence of lymphadenectomy, and positive surgical margins are inherent with this procedure. In some studies, enucleation has demonstrated similar 10-year overall and disease-free survival rates as formal resection [101]. Moreover, enucleation may be associated with shorter operative time, though no difference in postoperative outcomes has been demonstrated [101]. Additional studies are needed to elucidate the optimal management of small, nonfunctional PNETs, particularly focused on the appropriate size for active surveillance and operative approach for those necessitating resection.

Whether to perform formal oncologic resection of tumor-draining lymph node basins during surgery for PNETs similar to lymphadenectomy with other solid tumors remains unclear. Lymph node metastases have been correlated with worse overall survival in patients with PNETs, thus the value in performing lymphadenectomy at the time of primary resection remains an area of active investigation [102,103]. In a small retrospective study inclusive of 206 patients, lymph node metastasis was associated with worse overall survival, especially in patients with grade 2–3 disease [102]. Similarly, another small, single-institution retrospective study by Hasim et al. demonstrated lower median overall survival in patients with nodal metastases [103]. A recent 18-year retrospective analysis of a prospectively collected database of 314 patients with resected PNETs also suggests significant prognostic information from lymphadenectomy [101]. Of note, some studies have demonstrated that lymph node metastases are not correlated with oncologic outcomes [104] and also that lymphadenectomy may not affect overall survival in patients who undergo primary resection [105,106]. At present, consensus guidelines from the ENETs, Americas Hepato-Pancreato-Biliary Association (AHPBA), NCCN, and NANETS all include a recommendation or consideration for lymphadenectomy with PNET resection [8].

### 3.3. Genetic Disorders

Several genetic disorders predispose to the development of PNETs, and surgical management of PNET may be specific to these conditions. In these conditions, expectant management rather than aggressive surgical resection of small PNETs is established practice [107,108]. Von Hippel-Lindau (VHL) is an autosomal dominant inherited disease associated with increased risk of several tumors, including PNETs in up to 17% of all VHL patients. PNETs are less likely to be high grade in VHL patients. Additionally, given the associated risk of pancreatic cystic disease, pancreatic preservation is favored if surgery is required [109]. A nonoperative approach can be pursued more liberally in this population.

Multiple endocrine neoplasia (MEN) predisposes patients to tumors of the endocrine system, including PNETs. The mean age of death for MEN1 patients is 55 years old, with malignant PNETs comprising the primary cause of death [110]. For small PNETs associated with MEN1, including nonfunctional and gastrinomas, observation in these patients has recommended by several PNET societies [111]. Studies have shown there is no increased mortality for MEN1 patients with small nonfunctional PNETs (less than 2 cm) compared to MEN1 patients without PNETs [112]. A recent systematic review also confirmed that MEN1 patients with PNETs less than 2 cm can safely undergo active surveillance [113]. For tumors grade 2 or above, or greater than 2 cm, surgical resection should be pursued [113]. No overall or disease-free survival benefit is seen in patients who undergoing subtotal or total pancreatectomy [114], and a pancreas-preserving approach is preferred given the multifocality of tumors seen in MEN1.

### 3.4. Metastatic Disease

The most common anatomic location of metastasis from PNET is the liver. Palliative resection may be indicated to alleviate functional syndromes secondary to the hormone secretion, symptoms due to mass effect, or even for survival benefit [115,116,117,118,119,120]. In one study of patients with stage IV PNET, five-year overall survival has been shown to be improved to 56.6% compared to 23.9% for patients managed non-operatively [115]. These results were similar to a recent study using the surveillance, epidemiology, and end (SEER) database, which found that primary resection in patients with metastatic PNETs to the liver had improved overall survival at 5 years to 67.9% compared to 22.3% for patients who did not undergo resection [117]. The benefit of resection of the primary tumor and liver metastases is demonstrated even for more extensive pancreatic surgery such as pancreaticoduodenectomy with metastasectomy [118]. More controversial is the management of the primary pancreatic tumor, particularly in the setting of non-resected liver metastases. Resection of the primary pancreatic tumor in the setting of unresectable metastases is supported by a recent multi-centered retrospective study; in this study, patients with stage IV PNET who underwent primary tumor resection were found to have a 5-year survival rate of 65.4% compared to non-operative management of 47.8% [121].

Surgical metastasectomy for PNET must be weighed against perioperative morbidity and resection and high likelihood of recurrence. Recurrence of metastatic disease after liver-directed treatment has been noted to be as high as 94% over 5 years [122]. Additional liver-directed therapies may be indicated, offering potentially less morbidity than surgical metastasectomy but also extending liver-directed treatment to patients with disease that may be too disseminated for resection. Radiofrequency ablation, hepatic artery embolization, and radioembolization all may have a role for PNET liver metastases [123,124,125]. Radiofrequency ablation utilizes high-frequency current to induce coagulative necrosis and often is an adjunct to primary resection. Hepatic artery embolization and trans-arterial chemoembolization both implement catheter-guided placement of material to provide embolic occlusion of the hepatic artery, with the latter also instilling local chemotherapy to tumor tissue. Radioembolization is catheter-directed instillation of radioactive material within the tumor tissue and blood supply to the tumor. Per NANETS guidelines, these techniques are best reserved for patients who are not surgical candidates [126].

In one of the largest single-institution studies of PNET liver metastases, comparison of liver-directed treatment for neuroendocrine patients with liver metastases found hepatic resection of metastases is associated with a median survival of 160 months [119]. The median overall survival was significantly improved for patients who underwent hepatic resection compared to radiofrequency ablation (123 months), systemic therapy (70 months), chemoembolization (66 months), or observation (38 months) [119]. While this study was prospective and limited by nonrandomization, the results highlighted the efficacy and survival of liver-directed therapy. Particularly for PNET patients with liver metastases who are not surgical candidates, intervention on liver metastases alone is a treatment option.

In patients who are not surgical candidates, there are several liver-directed modalities that have been studied to decrease metastatic burden and provide symptom relief. Hepatic arterial embolization, chemoembolization, and radioembolization are other alternative treatment modalities that can provide palliative benefit in patients who are not surgical candidates. Embolization of the hepatic artery serves to decrease blood supply to malignant cells, as opposed to healthy hepatocytes that obtain the majority of their blood supply from the portal system. Several studies have demonstrated that embolization can result in symptomatic relief and provide survival benefit [127,128]. Progression-free survival has been reported to be approximately 19 months for hepatic artery embolization and chemoembolization [127]. No study has directly compared embolization modalities and differences in progression-free or overall survival [128].

Ablation, including radiofrequency ablation (RFA), cryoablation, and microwave ablation, is a modality that has been used to address PNET liver metastases as both an adjunct to surgical resection and as a primary treatment [122,129]. While data are limited, the largest study, which included 63 patients with neuroendocrine hepatic metastases, found that RFA can provide local control, in addition to symptomatic relief [129]. Data more specifically on cryoablation and microwave ablation are limited to small case series, though they may be a potential palliative option for metastatic disease [130,131,132,133]. Furthermore, selective internal radiation therapy (SIRT) has demonstrated potential efficacy in other liver metastases, namely in the setting of colorectal cancer [134]. SIRT appears to be well tolerated and alleviates symptoms in patients with liver metastases from primary PNETs; however, at present the data remains limited to small case series [135,136].

While there is data to suggest that liver transplantation for management of PNET liver metastases is feasible and provides survival benefit, transplantation for PNET patients remains controversial. Overall survival at 5 years after a liver transplantation in PNET patients with liver metastases has been reported as high as 80% [137]. Survival rates for PNET patients with liver metastases who undergo liver transplantation are similar to patients who undergo transplantation for hepatocellular carcinoma [138]. Select groups that have demonstrated the most benefit from liver transplantation include patients who are less than 55 years of age, have disease that is confined to the liver without extrahepatic extension, and with well-differentiated pathology [137,138,139]. However, defined patient selection criteria remain an area of ongoing investigation [125]. Additionally, given the already limited availability of liver donors, liver transplantation has yet to be a durable option for the treatment of liver metastases in PNET patients.

## 4. New Directions in the Treatment of Pancreatic Neuroendocrine Tumors

Historically, medical management of PNETs included somatostatin analogs for symptomatic treatment with adjuvant chemotherapy for symptomatic or metastatic tumors. The treatment landscape for locally advanced and metastatic PNETs continues to evolve. In locally advanced PNETs, 5-year overall survival is reported to be 91% with resection [140]. In a small retrospective study, neoadjuvant chemotherapy for patients with locally advanced or metastatic PNET disease demonstrated partial radiographic response in 43% of patients. A total of 87% of patients in that study who received neoadjuvant chemotherapy proceeded to surgical resection [141]. Additionally, several case studies have demonstrated the efficacy of neoadjuvant chemotherapy for locally advanced disease, with patients who received neoadjuvant therapy progressing to surgical resection [142,143]. These findings encourage comparison to treatment paradigms of pancreatic adenocarcinoma, where neoadjuvant chemotherapy allows for more patients to proceed to the operating room [144,145]. Even in the setting of metastatic disease, preoperative chemotherapy for patients with liver metastases has demonstrated improved median overall survival at 97.3 months compared to surgery alone at 65.0 months [146]. These studies highlight the importance of reducing tumor burden and the benefit of surgical resection on overall survival for patients with PNETs.

Targeted therapies have also been investigated in the treatment of PNETs. Sunitinib, a tyrosine kinase inhibitor, as an adjuvant therapy in a phase 3 clinical trial demonstrated a prolonged median progression-free survival at 11.4 months compared to 5.5 months in patients with well-differentiated PNETs [147]. The OPALINE study, a clinical trial examining the efficacy of sunitinib and/or everolimus, an mTOR inhibitor, in progressive unresectable or metastatic PNET, found overall survival in patients who received either or both treatments was comparable to standard chemotherapy regimens [148]. Additionally, in another phase 3 clinical trial, progression-free survival was increased to 11 months from 3.9 with the addition of everolimus in patients with advanced nonfunctional neuroendocrine tumors, including a pancreas origin [3].

Immunotherapy has been paradigm-shifting in the treatment of some malignancies. Targeting the PD-1 and PD-L1 pathways has become an area of active investigation for a range of tumor types. There have been promising results in advanced melanoma [149], triple-negative breast cancer [150], and rectal cancer [151]. The anti-PD-1 antibody pembrolizumab has been studied in PNETs. In a recent phase 2 clinical trial, three patients with PNETs who received pembrolizumab demonstrated a partial response [152]. While the patient number is small, this finding was notable in that response occurred in PD-L1 negative tumors, similar to observations in other PD-L1 negative solid tumors [153].

Given the increased expression of somatostatin receptor in PNETs, peptide receptor radionuclide therapy has been of recent interest as a targeted therapy strategy. In an international, multi-center phase 3 trial, 177Lu-Dotatate, a radiolabeled somatostatin analog, was compared to long-acting octreotide for patients with neuroendocrine tumors [154]. The 177Lu-Dotatate group had a median overall survival of 48 months compared to 36.3 months in the control group. While the survival difference was not statistically significant, this study demonstrates a clinically relevant difference and provides for a potential alternative treatment strategy for PNET patients who do not respond to somatostatin analogs.

## 5. Conclusions

Surgical management of pancreatic neuroendocrine tumors has evolved in the last several years. Localized functional PNETs require resection not only of curative-intent and prevention of metastases, but also for source control for hormone secretion and subsequent hormone-related symptoms. Data continue to support active surveillance rather than resection for select, incidentally identified nonfunctional PNETs. Increasing evidence suggests that resection of either the primary tumor or liver metastases can provide survival benefit in patients with metastatic PNET. For patients who are not surgical candidates, liver-directed therapies and systemic regimens remain viable options with demonstrable survival benefit for metastatic disease.

## Figures and Tables

**Figure 1 cancers-15-02006-f001:**
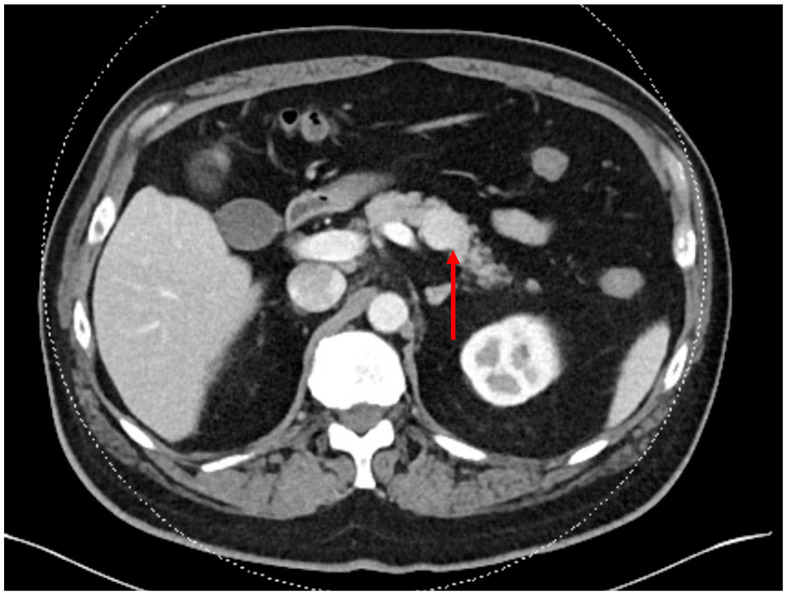
Contrast-enhanced CT demonstrating an incidental well-circumscribed 2.3 cm pancreatic mass (red arrow). The mass is enhanced relative to pancreatic parenchyma, consistent with a nonfunctioning pancreatic neuroendocrine tumor.

**Figure 2 cancers-15-02006-f002:**
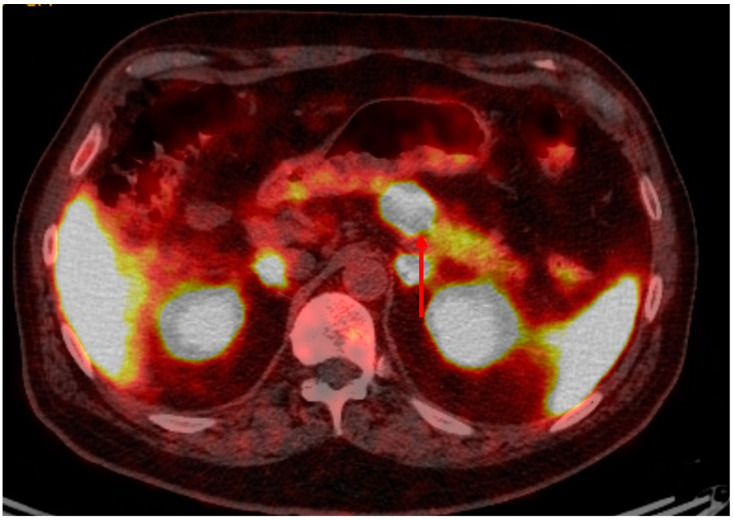
Gallium-68 Dotatate PET/CT (of the patient featured in Figure 1) demonstrating intense dotatate uptake in the pancreatic lesion and expected background activity, consistent with a somatostatin receptor positive/dotatate-avid neuroendocrine tumor.

**Figure 3 cancers-15-02006-f003:**
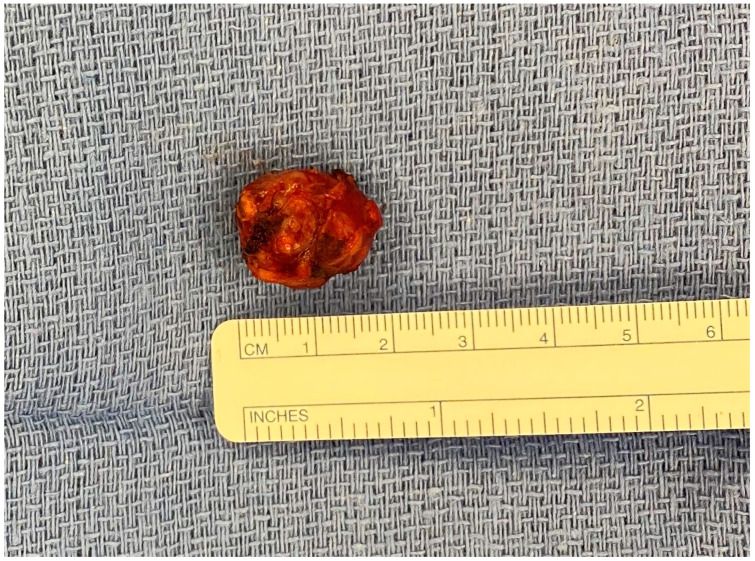
Enucleation specimen of a 2-cm insulinoma in the proximal pancreatic body without pancreatic duct involvement and localized on a preoperative MRI.

**Figure 4 cancers-15-02006-f004:**
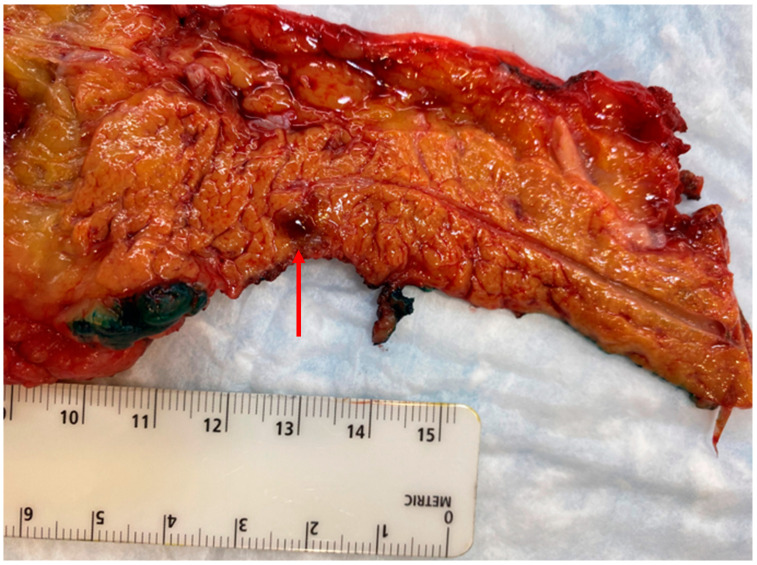
Distal pancreatectomy performed for a symptomatic, but diminutive (4 mm), insulinoma in the pancreatic tail that was not well visualized on cross-sectional imaging or endoscopic ultrasound, but localized with selective arterial calcium stimulation angiography.

**Table 1 cancers-15-02006-t001:** World Health Organization (WHO) classification of PNETs.

**Well-Differentiated**			
	**Ki67 Proliferation Index**	**Mitotic Index per High Power Field**	
Grade 1	<3%	<2 mitoses/2 mm^2^	
Grade 2	3–20%	2–20 mitoses/2 mm^2^	
Grade 3	>20%	>20 mitoses/2 mm^2^	
**Poorly Differentiated**			
	**Ki Proliferation Index**	**Mitotic Index per High Power Field**	**Cell Cytomorphology**
Small cell	>20%	>20 mitoses/2 mm^2^	Small
Large cell	>20%	>20 mitoses/2 mm^2^	Large

**Table 2 cancers-15-02006-t002:** Retrospective studies evaluating size cutoff value for nonfunctional PNETs’ active surveillance.

Study Authors	Year	Type of Study	Number of Patients (n)	Size Cutoff	Frequency of Serial Imaging	Outcome
Lee et al.	2012	Retrospective cohort, single institution	133 (56 surgery, 77 active surveillance)	<4 cm	Variable, CT or MRI	No difference in disease specific survival
Sadot et al.	2016	Retrospective cohort, single institution	181 (77 surgery, 104 active surveillance)	<3 cm	Variable	No difference in disease specific survival
Rosenberg et al.	2016	Retrospective cohort, single institution	35 (20 surgery, 15 active surveillance)	<2 cm	Every 6 months, CT or MRI	No difference in disease specific survival
Regenet et al.	2016	Retrospective cohort, multi-institution	80 (66 surgery, 14 active surveillance)	<1.7 cm	Variable	No difference in disease free survival
Kurita et al.	2020	Retrospective cohort, single institution	75 (52 surgery, 23 active surveillance)	≤2 cm	Every 6 months, CT and EUS (for first 5 years)	No difference in overall survival
Barenboim et al.	2020	Retrospective cohort, single institution	99 (55 surgery, 44 active surveillance)	<2 cm	Every 6 months, CT; Every 12 months, Gallium [67]. DOTATOC-PET	No difference in disease specific survival
Arra et al.	2022	Retrospective cohort, single institution	64 (41 surgery, 23 active surveillance)	<2 cm	Every 6 months, CT or MRI (for first 2 years)	No difference in disease progression rate

**Table 3 cancers-15-02006-t003:** Comparison of society recommendations for management of small nonfunctional PNETs.

	North American Neuroendocrine Tumor Society (NANTS)	National Comprehensive Cancer Network (NCCN)	Canadian National Expert Group	European Neuroendocrine Tumor Society (ENETS)
Active Surveillance	Size 1 cm or less Asymptomatic	Size less than 1 cm preferred, but can selectively be observed if less than 2 cm Asymptomatic Low-grade Incidentally discovered	Size less than 2 cm Solitary lesion with no evidence of invasive disease Low Ki67 Stability on serial surveillance	Size less than or equal to 2 cm Asymptomatic Low-grade No evidence of malignant potential
Consideration of Resection	Tumors 1–2 cm *	Invasive Node-positive tumors	Progression on surveillance	Symptomatic Higher grade (G2) Patient preference

* per NANTS guidelines, for nonfunctional PNETs that are 1–2 cm, surgical rection can be considered pending patient comorbidities, tumor grade, extent of resection required, patient preference, and access to follow-up care.

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
