# Peer review of "Surgical Management of Pancreatic Neuroendocrine Tumors"

_cancers, 2023, doi:10.3390/cancers15072006_

Round 1

Reviewer 1 Report

This is an interesting review on surgical approach to pancreatic neuroendocrine tumors. As such, it helps clarify certain contradictory findings in the literature and deserves publication.

Here lists a few of my concerns that could improve the overall quality of the manuscript if addressed appropriately.

In my opinion, the manuscript could be improved, if the following data are added:

1.     Comments on rare / very rare functional PNETs that secrete a variety of hormones or biogenic amines,  growth hormone-releasing hormone (GHRH), ACTH, serotonin, tachykinins,  PTH-related peptide, renin,  luteinizing hormone, cholecystokinin (CCK), erythropoietin, insulin-like growth factor-2 (IGF-2), or glucagon-like peptide-1 (GLP-1)

2.     data on endoscopic ultrasound (EUS), Intraductal ultrasound (IDUS), Intraoperative ultrasound (IOUS), since EUS is currently considered to be the most sensitive method of detecting pancreatic neuroendocrine tumours, especially small ones with diameter < 2 cm

3.      comments on palliative treatment recommended in patients for whom resection of liver metastases is not possible (including selective hepatic artery embolization (HAE), transarterial chemoembolization (TACE), transarterial radioembolization (TARE), selective internal radiation therapy (SIRT)],  RFA, cryoablation, and microwave ablation (MWA) as well as irreversible electroporation (IRE)

4.     comments on liver transplantation that is performed in selected group of patients with severe symptoms related to the production of hormones or amines (patients under 60 years of age, without extrahepatic metastases, and with low expression of Ki-6may benefit from transplantation).

Author Response

March 23, 2023

Re: Manuscript ID: cancers-2272495
Type of manuscript: Review
Title: Surgical Approach To Pancreatic Neuroendocrine Tumors
Authors: Megan L Sulciner, Thomas E. Clancy *

To members of the editorial board,

We are grateful for consideration of our invited review for publication and appreciate the comments and suggestions of our reviewers.

We have attempted to address these thoroughly and have added text and references to the manuscript as needed. Our responses to the reviewer inquires are summarized below.

Thank you for your consideration.

Thomas Clancy

(corresponding author)

Reviewer 1

This is an interesting review on surgical approach to pancreatic neuroendocrine tumors. As such, it helps clarify certain contradictory findings in the literature and deserves publication.

Here lists a few of my concerns that could improve the overall quality of the manuscript if addressed appropriately.

In my opinion, the manuscript could be improved, if the following data are added:

  1. Comments on rare / very rare functional PNETs that secrete a variety of hormones or biogenic aminesgrowth hormone-releasing hormone (GHRH), ACTH, serotonin, tachykinins,  PTH-related peptide, renin,  luteinizing hormone, cholecystokinin (CCK), erythropoietin, insulin-like growth factor-2 (IGF-2), or glucagon-like peptide-1 (GLP-1)

Thank you for this suggestion. We agree that additional discussion of very rare PNETs should be included. We have now added commentary on very rare PNETs within the “Presentation and diagnosis of pancreatic neuroendocrine tumors” section, paragraphs 10-12.

  1. data on endoscopic ultrasound (EUS), Intraductal ultrasound (IDUS), Intraoperative ultrasound (IOUS), since EUS is currently considered to be the most sensitive method of detecting pancreatic neuroendocrine tumours, especially small ones with diameter < 2 cm

We have now added under “Presentation and diagnosis of pancreatic neuroendocrine tumors” section, paragraphs 4-5, additional discussion on the current data regarding the sensitivity and utility of EUS, IDUS and intraoperative ultrasound.

  1. comments on palliative treatment recommended in patients for whom resection of liver metastases is not possible (including selective hepatic artery embolization (HAE), transarterial chemoembolization (TACE), transarterial radioembolization (TARE), selective internal radiation therapy (SIRT)],  RFA, cryoablation, and microwave ablation (MWA) as well as irreversible electroporation (IRE)

Thank you for this suggestion. We agree these are important modalities to discuss in the management of liver metastasis. We now discuss the implications and benefits of these therapies in under the “Metastatic disease” section, paragraphs 4-5.

  1. comments on liver transplantation that is performed in a selected group of patients with severe symptoms related to the production of hormones or amines (patients under 60 years of age, without extrahepatic metastases, and with low expression of Ki-67 may benefit from transplantation).

We have added additional commentary on specific patient populations that may benefit from liver transplantation in the “Metastatic disease” section, paragraphs 6.

Reviewer 2 Report

The review "Surgical approach to pancreatic neuroendocrine tumors" is well written and gives a very nice and usefull overview of the field and i deserves publication

A few comments

 Authors state:

The management of small nonfunctioning tumors has been an area of debate given their typically relatively indolent behavior. While additional tumor characteristics such as Ki-67 proliferative index could ideally guide decision making regarding resection.

 Comment 1:  It would be nice with a recommendation of a specific KI67 index in the text

 Importantly, all patients followed nonoperatively require definitive diagnosis via either fine needle aspiration (FNA) or somatostatin-receptor imaging.

Comment 2: Sometimes it is not possible to do a biopsy because the tumor is small. Further it might not always be advisable/necessary.In these cases, when would you do a follow up scan so you dont miss that it is an aggresive tumor. Further if the biopsy is not possible how do you rule out that it isnt spleen tissue. I think this should be mentioned.

The nonoperative strategy is beneficial primarily to avoid the known incidence of morbidity from surgical exploration of up to 62%, mainly due to pancreatic fistula.

Comment 3: the patients (especially after a Whipple) suffer from decreased quality of life (for life) with SIBO, exocrine insufficiency, pain etc. I think this should be briefly mentioned that it is a considerable risk.

comment 4: The use of robotic surgery is increasing. Benefits and the use should be mentioned. 

references that could be added to guide decision making with regards to tumor size:

1:Beltini R: 2011: Surgey

2:Krogh S: 2022: Frontiers in endocrinology

Author Response

March 23, 2023

Re: Manuscript ID: cancers-2272495
Type of manuscript: Review
Title: Surgical Approach To Pancreatic Neuroendocrine Tumors
Authors: Megan L Sulciner, Thomas E. Clancy *

To members of the editorial board,

We are grateful for consideration of our invited review for publication and appreciate the comments and suggestions of our reviewers.

We have attempted to address these thoroughly and have added text and references to the manuscript as needed. Our responses to the reviewer inquires are summarized below.

Thank you for your consideration.

Thomas Clancy

(corresponding author)

Reviewer 2

The review "Surgical approach to pancreatic neuroendocrine tumors" is well written and gives a very nice and useful overview of the field and it deserves publication. 

A few comments

Authors state: The management of small nonfunctioning tumors has been an area of debate given their typically relatively indolent behavior. While additional tumor characteristics such as Ki-67 proliferative index could ideally guide decision making regarding resection.

 Comment 1:  It would be nice with a recommendation of a specific KI67 index in the text. Importantly, all patients followed nonoperatively require definitive diagnosis via either fine needle aspiration (FNA) or somatostatin-receptor imaging.

Thank you for this suggestion. We highlight the WHO recommendations for specific Ki67 index under “Introduction”, paragraph 2 and in Table 1. We have now added additional commentary and data regarding tissue biopsy specifically and EUS FNA as a modality in “Introduction”, paragraph 3.

Comment 2: Sometimes it is not possible to do a biopsy because the tumor is small. Further it might not always be advisable/necessary. In these cases, when would you do a follow up scan so you don’t miss that it is an aggressive tumor. Further if the biopsy is not possible how do you rule out that it isn’t spleen tissue. I think this should be mentioned. The nonoperative strategy is beneficial primarily to avoid the known incidence of morbidity from surgical exploration of up to 62%, mainly due to pancreatic fistula.

 We have added additional data regarding the rate of adequate biopsy in smaller tumors under “Introduction”, paragraph 3. In regards to the nonoperative strategy and morbidity associated with pancreatic resection, please see our response in Comment 3.

Comment 3: The patients (especially after a Whipple) suffer from decreased quality of life (for life) with SIBO, exocrine insufficiency, pain etc. I think this should be briefly mentioned that it is a considerable risk.

We agree this is an important factor in the decision-making process of pursuing active surveillance. We have added data on the rates of complications following pancreatic resection under “Nonfunctional PNETs”, paragraph 6.

Comment 4: The use of robotic surgery is increasing. Benefits and the use should be mentioned. References that could be added to guide decision making with regards to tumor size:

1:Beltini R: 2011: Surgey

2:Krogh S: 2022: Frontiers in endocrinology

Thank you for this suggestion. Under “Nonfunctional PNETs”, paragraph 7 we have no added additional discussion on the role of minimally invasive techniques, including robotic surgery approaches for PNETs. Additionally, we have added Bettini et al. and Krogh et al. as references in our discussion on approach to small nonfunctional PNETs under “Nonfunctional PNETs”, paragraph 4. 

Round 2

Reviewer 1 Report

Dear Authors,

Thank you for the corrections. Congratulations.